# Pumpkin Oil and Its Effect on the Quality of Naples-Style Salami Produced from Buffalo Meat

**DOI:** 10.3390/foods14061077

**Published:** 2025-03-20

**Authors:** Francesca Coppola, Filomena Nazzaro, Florinda Fratianni, Silvia Jane Lombardi, Luigi Grazia, Raffaele Coppola, Patrizio Tremonte

**Affiliations:** 1Department of Agricultural Sciences, University of Naples “Federico II”, Piazza Carlo di Borbone 1, 80055 Portici, Italy; francesca.coppola2@unina.it; 2Institute of Food Science, Italian National Research Council, Via Roma 64, 83100 Avellino, Italy; filomena.nazzaro@cnr.it (F.N.); fratianni@isa.cnr.it (F.F.); 3Department of Agricultural, Environmental and Food Sciences (DiAAA), University of Molise, Via De Sanctis snc, 86100 Campobasso, Italy; coppola@unimol.it (R.C.); tremonte@unimol.it (P.T.); 4Department of Agricultural and Food Sciences, Alma Mater Studiorum, University of Bologna, Piazza Goidanich 60, 47521 Cesena, Italy

**Keywords:** *Lactilactobacillus sakei*, *Staphylococcus xylosus*, low acidification dry salami, meat oxidation, *Enterobacteriaceae*

## Abstract

The use of buffalo meat in fermented sausage production represents a sustainable and innovative approach to enhancing the value of underutilized meat cuts. However, its high heme content and specific fatty acid composition makes the meat particularly sensitive to lactic fermentation with lipid oxidation phenomena and sensory character decay. Therefore, buffalo meat requires tailored fermentation strategies to ensure product stability. The aim of this study was to optimize fermentation strategies by exploring milder acidification processes and the fortification of buffalo meat with vegetable oils to reduce oxidation while maintaining microbiological quality. In particular, the effect of adding or omitting glucose and fortifying with pumpkin seed oil in Napoli-style buffalo salami was studied and the impact on the main quality parameters was evaluated. Pumpkin seed oil (0.5%) was selected for its antimicrobial and antioxidant properties and evaluated for its interaction with starter cultures through Minimum Inhibitory Concentration (MIC) tests and predictive microbiology models. Based on the findings, its use was validated in Napoli-style salami, produced with and without glucose. Microbial dynamics, physicochemical changes over time, oxidation indices, and sensory attributes were assessed. Results indicated that the sugar-free formulations combined with pumpkin seed oil achieved optimal sensory and safety attributes. The addition of glucose facilitated rapid lactic acid bacterial growth (about 2.5 ∆ log CFU/g), enabling pH reduction to safe levels (<5.2) and the effective inhibition of *Enterobacteriaceae* and coliforms. However, acidification in the control batch, as demonstrated by multiple variable regression analyses, induced pre-oxidative conditions, increasing lipid oxidation markers (TBARSs > 0.7 mg MAD/Kg), which negatively impacted flavor and color stability. The use of pumpkin seed oil confirmed its antimicrobial and antioxidant potential, making it a promising fortifying ingredient for producing slow-fermented, mildly acidified (pH > 5.4) buffalo meat salami, offering a novel strategy for improving the nutritional, sensorial, and safety quality of dry fermented meat.

## 1. Introduction

Red meat and its products are good sources of nutrients, including high-quality protein (essential and non-essential amino acids), fatty acids, and different micronutrients like iron, zinc, selenium, and vitamins D and B12 [1,2,3]. However, many chronic diseases, such as diabetes, cancer, cardiovascular diseases, and obesity, are also attributed to the excessive consumption of these products. The risk factors are mainly their saturated fatty acid and cholesterol content [4,5]. Despite this, it is still important to eat a moderate amount of meat as it is a key strategy for obtaining essential nutrients. On the other hand, meat products lend themselves to the addition of various bioactive compounds for the development of functional foods and to counteract the negative effects of excessive consumption [6,7,8,9]. A review by Espinales et al. [10] provides scientific information on appropriate strategies to improve meat products’ nutritional profile and health benefits, highlighting the importance of reducing saturated fat, sodium, and cholesterol and incorporating functional ingredients (fiber, antioxidants, and omega-3 fatty acids). Niu et al. [11] underlines the importance of using natural additives to delay protein oxidation and enhance minced mutton’s shelf life and quality during chilled storage. Many natural substances combine their antioxidant power with a strong activity against pathogenic or altering microorganisms in fresh and preserved food products [12,13,14,15,16,17]. In this context, there is also growing interest in diversifying traditional meat products to suit evolving consumer preferences for healthier, more sustainable, and innovative food options [18,19]. Among such alternatives, buffalo meat is gaining attention due to its nutritional profile, which includes a lower fat content and a higher proportion of beneficial nutrients, like protein, iron, and omega-3 fatty acids [20]. This type of meat is widely used in many parts of the world, especially for the preparation of fresh products, and its consumption is widespread and is often connected with ethical and religious customs [21,22,23]. Buffalo meat is an excellent source of high-quality proteins comparable to beef. Its proteins are rich in essential amino acids, which are important for muscle repair and general body functions. The fat content of buffalo meat is generally lower than that of beef. The fat content (with a lower percentage of saturated fat than meat from other species) generally varies between 5 and 10%, depending on the specific diet and portion of buffalo consumed. Buffalo meat is a good source of iron, particularly hem iron, which is more readily absorbed by the body than vegetable iron. It is also a good source of zinc, phosphorus, and B vitamins, including niacin and riboflavin. These are essential for energy metabolism, skin health, and the proper functioning of the nervous system [24]. When derived from young animals, consumers generally appreciate buffalo meat for its taste and texture [25]. Tamburrano et al. [20] showed that buffalo meat is a viable alternative to other types of meat from a health perspective, particularly for individuals with specific physiological conditions, such as pregnancy, and those at risk of cardiovascular and cerebrovascular diseases. Bioactive compounds from plant-derived ingredients, including carrot, pumpkin peel, and apple pomace, have been used to supplement the composition of buffalo meat products, as widely reported in the literature [26,27,28,29,30].

However, a challenge in the use of buffalo meat in sausage production is its relatively lean nature, which can negatively affect the texture and palatability of the final product. In this direction, different ingredients to be incorporated into the sausage matrix have been studied; from the use of buffalo and pork meat mixtures [31] to the use of oils from various plant matrices [32]. In particular, the incorporation of vegetable oils during salami processing is a practice that has been gaining increasing interest, particularly for its potential to enhance the nutritional and functional properties of the product. The advantages of adding vegetable oils to salami are the improvement of the lipid profile, as they are rich in unsaturated fatty acids, which can help reduce the amount of saturated fats in the final product, making it a healthier option; some oils, such as pumpkin seed oil and olive oil, contain bioactive compounds (polyphenols, tocopherols, and carotenoids) that can limit lipid oxidation and extend the shelf life of the salami. Therefore, the use of buffalo meat in fermented products enriched with various ingredients seems to be a suitable strategy. Fermented meat products, widely used throughout the world, are the result of a complex fermentation process in which microbial cultures, including lactic acid bacteria, are used to acidify the mixture and inhibit the growth of harmful organisms, contributing to the development of unique sensory characteristics [33,34,35]. The elements of typicality and connection with the territory strongly influence the propensity to consume this type of food [36]. Based on all these considerations, Naples-style salami, a cornerstone of the delicatessens of the geographical area with a strong buffalo industry, would represent a valid model for the use of buffalo meat and the experimentation of new ingredients and fermentation processes. Specifically, Napoli-type salami is produced in the Campania region, the cradle of buffalo breeding and production. While the most prized parts of the buffalo easily find a market, the less renowned cuts, such as the shoulder, could be used in the innovative production of traditional cured meats, thus contributing to the preservation and renewal of local gastronomic traditions, with a positive impact on the economy and the sustainability of production. This type of salami is characterized by a finely balanced mixture of meat, fat, and other ingredients, subjected to a careful fermentation and drying process [37,38,39]. A study by Fratianni et al. [40] found that pumpkin seed oil possesses strong antioxidants and anti-inflammatory and antibacterial properties, making it a candidate for potential applications in the food industry. In lean meat-based products (such as those made with buffalo meat or game meat), the addition of vegetable oils can compensate for the lower amount of animal fat, improving the softness and juiciness of the product.

Based on all these considerations, the aim of our study was to evaluate the possibility of formulating a Naples-style salami containing buffalo meat and incorporated pumpkin seed oil as a functional ingredient and for its health benefits, and to obtain a final product with the intact sensory qualities of the original product. This research evaluated the impact of buffalo meat on the salami’s flavor and color, as well as the effects of pumpkin oil on its microbiological safety, shelf life, and overall consumer acceptability. By exploring the intersection of tradition and innovation, this paper seeks to contribute to the growing body of knowledge on the development of alternative meat products and functional cured meats that meet contemporary demands for health-conscious and sustainable food options.

## 2. Materials and Methods

### 2.1. Vegetable Oil

Pumpkin seed oil (Merck KGaA, Darmstadt, Germany), in liquid form and light yellow in color, was chosen for its biological activities already studied by this research group. Specifically, the oil was chosen for its antioxidant and antimicrobial activity against unwanted gram negatives such as *Escherichia coli* and *Pseudomonas aeruginosa*, and against gram positives, such as *Listeria monocytogenes* and *Staphylococcus aureus*. In the present study, the Minimal Inhibitory Concentration (MIC) of pumpkin oil on the growth of *Escherichia coli* was assessed using sterile Luria Broth with the resazurin test, followed by 24 h incubation at 37 °C. The MIC was evaluated by observing color changes [40].

### 2.2. Microorganisms

*Latilactobacillus sakei* 152 (*Lt. sakei* 152) and *Staphylococcus xylosus* MVS9 (*S. xylosus* MVS9) were used as starter cultures. Both strains, belonging to the collection of the Department of Agricultural, Environmental and Food Sciences (University of Molise), were previously selected for their fermentative and technological characteristics [41].

#### Effect of Oil on Starter Culture Strains

The effect of pumpkin oil on the growth of strains used as starters was evaluated through the determination of Minimal Inhibitory Concentration (MIC), using the resazurin microtiter plate assay [42,43]. Multiwell plates were prepared in triplicate and incubated at 37 °C for 24 h. Additionally, the impact of pumpkin seed oil on the growth of starter culture strains was assessed in a sarcoplasmic protein extract (SPE) prepared following the method of Tremonte et al. [41], with slight modifications. Briefly, buffalo *Longissimus dorsi* muscle samples were suspended in water (1:20 *w*/*v*), homogenized (PBI International, Italy) for 5 min at 0 °C (ice bath), and centrifuged at 1200× *g* for 30 min at 4 °C (Eppendorf Centrifuge 570 R G). The resulting supernatant was filtered (0.45 µm, Millipore, Ireland) to obtain a protein concentration of 10 mg/mL. The SPE was supplemented with glucose (10 mg/mL), inoculated with actively growing indicator strains, combined with 0.5% pumpkin seed oil, and incubated at 28 °C for 30 h. In all cases, the inoculum of growing cells was adjusted to obtain an OD 600 nm value of 0.2 (corresponding to about 4 Log CFU/mL). Microbial growth of *Lt. sakei* 152 and *S. xylosus* MVS9 incubated at 28 °C was monitored for 30 h by plate counts performed in MRS agar (Oxoid, Milan, Italy) and Mannitol Salt Agar—MSA—(Oxoid, Milan, Italy), respectively, and growth kinetic parameters were calculated using the structured dynamic model of Baranyi and Roberts [44]. Growth kinetic parameters were compared with those found under culture conditions that differed in the absence of pumpkin seed oil.

### 2.3. Effectiveness in Fermented Sausage

The antimicrobial efficacy of pumpkin seed oil and its compatibility with fermentation processes was validated in a Napoli-style fermented salami made from buffalo meat. The amount of pumpkin seed oil to be used for salami production was chosen based on MIC results against undesirable and useful microorganisms.

#### 2.3.1. Fermented Sausage Preparation

The lean meat of male Mediterranean buffalo (*Bubalus bubalis*), about 400–450 days old, and pork fat were used to produce Naples-type salami [45]. In particular, the finest cuts of the boneless shoulder (97%), such as chuck and brisket, were chosen. The meat and lard were minced in a meat grinder with 6 mm (diameter) holes. The minced mix was combined with NaCl (2.2%), black pepper (0.02%), KNO_2_ (0.008%), wild fennel (0.01), and garlic (0.03%) and was crushed in wine and strained. Ingredients, additives and spices were all purchased from Fratelli Pagani S.p.A. (Milan, Italy). Finally, the starter culture, containing *Latilactobacillus sakei* (10E11 CFU/100 kg) and *Staphylococcus xylosus* (10E11 CFU/100 kg), was added to obtain about 6.5 Log CFU/g. The preparation was divided into four batches:G20, glucose (0.2%) and pork lard (3%) were added to the mixture, and the batch was used as a control.G20P, produced with the addition of glucose (0.2%), pork lard (2.5%), and pumpkin oil (0.5%) sprayed during the second mixing.G0, produced without glucose addition, and pork lard (3%) was added to the mixture.G0P, produced without glucose addition, pork lard (2.5%) and pumpkin oil (0.5%) sprayed during the second mixing.

Each batch was stuffed into natural pork casings to ensure uniform dimensions (20 cm length, 5 cm diameter). After filling, the salami underwent a stabilization phase at 2–4 °C for 12 h. Initial drying was conducted for seven days, during which the relative humidity gradually increased from 60% to 80%, while the temperature gradually reduced from 21 °C to 11 °C. Thereafter, the ripening phase continued until the 49th day under controlled conditions (80% relative humidity and 11 °C temperature).

For all analytical determinations, two different samples of each batch were collected. The samples were transported to the laboratory under refrigerated conditions and analyzed within two hours after sampling. The analyses were carried out in triplicate at 0, 3, 7, 14, 21, 28, 35, and 49 days of ripening.

#### 2.3.2. Chemical Composition

The centesimal composition of the samples, including moisture, carbohydrate, protein, fat, and ash, was analyzed according to official AOAC methods [46]. Specifically, for moisture content determination, 5 g of dry salami was treated at 105 °C for 16 h, calculating the weight difference before and after drying. Non-Protein Nitrogen (NPN) was determined in accordance with AOAC method 973.31. Protein concentration was evaluated by the Kjeldahl method, while crude fat content was quantified by Soxhlet in accordance with AOAC method 960.39. For ash determination, the sample was treated at 550 °C for three hours. The carbohydrate content (C) was calculated according to the following formula:C% = 100 − (M + P + L + A + F)
where M = moisture (%); P = protein content (%); L = lipid content (%); and A = ash content (%).

The use of high-resolution gas chromatography allowed us to evaluate the composition of fatty acids. From the transesterification of approximately 25 mg of each fat sample, previously dissolved in 2 mL of petroleum ether and 2 mL of BF3-methanol reagent, the methyl esters of fatty acids were determined. The determination of the concentration of saturated fatty acids (SFAs), monounsaturated fatty acids (MUFAs), and polyunsaturated fatty acids (PUFAs) was performed as described by Tremonte et al. [47]. Results were expressed as percentage/dry matter (%/d.m.).

#### 2.3.3. Lipid Peroxidation

The quantification of thiobarbituric acid reactive substances (TBARSs) was performed to assess the extent of lipid oxidation, following a protocol adapted from previously established methods [47] using the thiobarbituric acid (TBA) assay, with trichloroacetic acid (TCA) employed as an extraction solvent. Briefly, 10 g of homogenized sample was distilled, and an aliquot of the distillate was reacted with an aqueous solution of 2-thiobarbituric acid (TBA). The mixture was heated in a boiling water bath for 35 min to allow chromophore formation, then rapidly cooled in an ice bath. The absorbance of the resulting TBA–malondialdehyde (MDA) complex was measured spectrophotometrically at 532 nm. TBARS values were calculated based on a standard curve generated using 1,1,3,3-tetraethoxypropane (TEP) as a reference compound. The results were expressed as micrograms of malondialdehyde (MDA) per gram of dry matter (µg MDA/g), providing a quantitative measure of lipid peroxidation in the samples.

#### 2.3.4. Physicochemical and Microbiological Analyses

The pH and activity of the water were evaluated, respectively, with a pH meter with lance probe (Mettler Toledo, Novate Milanese, Italy) and a Water Activity Meter CR2 (AQUALAB Instrument, Pullman, WA, USA). For microbiological analysis, 10 g of each sample was homogenized in peptone water (0.1%) using a Stomacher 400 (Seward Medical, London, UK) and subsequently subjected to decimal dilutions. Lactic bacteria (LAB), micrococci and staphylococci (Coagulase-Negative Cocci, CNC), and *Enterobacteriaceae* were enumerated under specific incubation conditions [41,47]. Five colonies were selected from the MRS and MSA plates and characterized by RAPD-PCR analysis [48,49] to assess the presence of initial cultures throughout the maturation period.

#### 2.3.5. Sensorial Characterization

Sensorial analysis was performed following the protocol of Chiavari et al. [50]. Twenty judges, aged between 22 and 65 and trained in the sensory analysis of salami, were used for the sensory evaluation. All participants have given their informed consent (Appendix A). In detail, 40 panelists, after 10 training sessions, formed a group of 20 judges who presented optimal uniformity of responses. Each judge was given a slice of salami 5 mm thick placed in a polyethylene container, which was hermetically sealed and presented individually and anonymously. The final evaluation was conducted through a descriptive analysis, assessing the perceived intensity of 13 different attributes according to a nine-point scale. The panelists, led by a panel leader, conducted a triple sensory analysis to evaluate smell and odor—in addition to intensity, the prevailing descriptor was reported, including those of the meat family (fresh meat, sour meat, fat, and aged meat), the animal family (stable, gut, and leather), the spicy family (green pepper, garlic, cinnamon, red pepper, and nutmeg), and other descriptors (such as acetic, oxidized, mold, rancid, and ammonia)—as well as to assess flavor (salty, acidic, and bitter), trigeminal sensations (spicy), color (evaluating intensity and uniformity), texture with reference to compactness, elasticity, hardness, moisture, and chewability. The panelists considered the specific evaluation criteria for each attribute as reported in Appendix A.

### 2.4. Statistical Analyses

Both data analysis and graph processing were carried out using GraphPad Prism 10.4.1 Approach. The data of the growth kinetics of the microorganisms in the model food substrate were subjected to the *t*-test to identify the differences and their significance levels (*p* < 0.05) between those of the microorganisms cultured in the presence of pumpkin seed oil and those without added pumpkin seed oil. The data on the development of microbial populations and chemical/physical parameters during the ripening period, as well as the scores of sensory attributes, were subjected to analysis of variance (ANOVA) followed by a Bonferroni post hoc test to identify differences and their significance levels (*p* < 0.05 or *p* < 0.01) between the different conditions. The influence of independent variables (acidification, LAB growth, and pumpkin seed oil use) on the level of TBARSs was assessed by means of multiple linear regression analysis.

## 3. Results and Discussion

### 3.1. Vegetal Oil as Alternative Ingredient

The MIC assessment confirmed the inhibitory effect of pumpkin seed oil against *Escherichia coli*, with bacterial growth being suppressed at an oil concentration starting from 46 µg/mL (±3). In contrast, the oil did not exhibit strong antimicrobial properties against the starter cultures, which grew even at concentrations exceeding 150 µg/mL. This finding reassured us about the potential of incorporating this seed oil as a functional ingredient in salami production. Although using natural compounds in fermented meat products is a promising approach, it remains relatively unexplored. Apricot seed oil and grape seed oil demonstrated broad antibacterial activity against *Staphylococcus aureus*, *Klebsiella pneumoniae*, *Pseudomonas aeruginosa*, and *E. coli*, positioning them as promising functional ingredients for application in fermented meat product formulations [51]. Similarly, Papatzimos et al. [52] suggested that the use of 2% hemp seed oil in fermented salami production could serve as a functional component to enhance fermented meat products’ nutritional profile and health benefits, particularly when combined with reduced sodium nitrite levels. To our knowledge, no research has been carried out to assess the influence of pumpkin seed oil on the growth of lactic acid bacteria. A pumpkin seed-based matrix was utilized to develop an alternative yogurt, which did not inhibit the proliferation of lactic acid bacteria. The viable cell counts of *Lactobacillus* spp. and *Bifidobacterium* spp. reached 7.17 log CFU/g and 7.82 log CFU/g, respectively [53].

#### Culture Starters—Pumpkin Oil Compatibility

The results of the Minimum Inhibitory Concentration (MIC) evaluation against the *Lt. sakei* 152 and *S. xylosus* MVS9 strains revealed a value of 150 µL/mL. Considering these results and previously reported MIC values for *Escherichia coli*, a concentration of 0.5 percent pumpkin seed oil was finally chosen for sausage preparation. To assess the influence of pumpkin seed oil on the growth of starter culture strains, the development of *Lt. sakei* 152 and *S. xylosus* MVS9 was monitored in a meat broth system, as a model food medium, with and without the addition of pumpkin seed oil (Appendix A). The kinetic parameters analyzed (Table 1), maximum growth rate, lag phase duration, and final cell concentration, showed no significant differences between the control and oil-fortified medium, suggesting that the presence of pumpkin seed oil at a concentration of 0.5 percent does not interfere with the growth dynamics of the two starter cultures.

Therefore, the incorporation of pumpkin seed oil does not impair the performance of the essential microbial strains responsible for fermentation. Consequently, its use could represent a viable strategy to improve the antimicrobial and antioxidant properties of the final product without adversely affecting the activity of the starter cultures. These findings align with previous research, highlighting the adaptability of specific lactic acid bacteria (LAB) and Coagulase-Negative staphylococci (CNS) to lipid-rich or polyphenol-rich environments [54,55,56]. The ability of these strains to thrive despite the presence of bioactive compounds further emphasizes the importance of carefully selecting starter cultures based on their resilience to environmental stressors. Several studies have demonstrated that polyphenols and unsaturated fatty acids can exert antimicrobial effects, potentially inhibiting bacterial growth [57]. However, the starter strains used in this study showed some robustness, probably due to several inherent resistance mechanisms, as already reported in the literature [58,59,60]. This underscores the need for strain-specific selection, ensuring that starter cultures possess the desired technological traits (e.g., acidification capacity and proteolytic activity) and the ability to withstand stressors present in complex food matrices [61,62]. Moreover, these findings reinforce the significance of isolating starter cultures from relevant ecological niches, as strains originating from specific meat environments tend to exhibit a superior adaptation to the physicochemical conditions of fermented meat products [63,64,65,66]. A meticulous characterization of starter cultures, including stress resistance and metabolic profiling, is essential for optimizing fermentation processes and ensuring product safety and sensory quality.

### 3.2. Effect of Pumpkin Oil on Sugar-Free Buffalo Salami Quality

The technological compatibility and antimicrobial efficacy of pumpkin oil were evaluated and validated in situ by monitoring the quality parameters of fermented buffalo sausages. The heterogeneous environmental conditions that characterize fermented sausages could influence and impair pumpkin oil’s effectiveness. Any pockets of residual air could generate oxidation phenomena of the polyunsaturated acid fraction of the vegetable oil, negatively affecting sensory and nutritional quality [67]. Oxidative events under normal conditions should be averted by the polyphenolic and antioxidant components of the oil itself. Still, polyphenols could interact through hydrogen and hydrophobic bonds with meat proteins in a complex system, such as fermented cured meats, decreasing availability and bioactivity [68]. For these and/or other interactions, the discrepancy between predictions based on in vitro data and actual in situ occurrences remains high. Indeed, despite the availability of extraordinary predictive tools [69], validation in model foods or on an industrial scale remains a must. In the present study, the efficacy of pumpkin oil (used with and without added sugars) was validated in an Italian-style fermented salami with buffalo meat.

#### 3.2.1. Effect on Chemical Composition and Fatty Acids

Table 2 shows the centesimal composition of the sausages at the end of maturation. The data show that the moisture and protein, fat, ash, and NPN content did not differ significantly between the batches. The effect of the use of vegetable oils on the centesimal composition is quite controversial [70,71]; some authors found significant effects on product moisture, as in the case of the use of chia oil, linseed, or olive oils [72], while others, such as Monteiro et al. [73], found no substantial differences. Our results evidence that the use of pumpkin oil as a partial replacement for pork fat did not change the quantitative chemical composition and consequently did not affect the evolution of the physicochemical and biochemical phenomena.

In contrast, as expected, significant differences were found in relation to fatty acid composition. Other authors have widely found an increase in monounsaturated and polyunsaturated fatty acids [74,75,76]. In our study, although pumpkin oil has only partially replaced pork fat, the concentration of MUFAs and PUFAs in batches with pumpkin oil is substantially higher than that found in batches prepared without vegetable oil. Consequently, the PUFA/SFA ratio, by the presence of pumpkin oil, increases from about 0.34 to about 0.70. The PUFA/SFA ratio is often considered to assess the effect of animal products on cardiovascular health [77,78] with reference to the impact on low-density lipoprotein cholesterol (LDL-C) [79].

#### 3.2.2. Effect on Microbial Populations and Physicochemical Parameters

The effect of pumpkin oil and sugar removal on the performance of virtuous and undesirable microorganisms and, more generally, on fermentation performance was evaluated. The behavior of the most relevant microorganisms and the pH trends are shown in Figure 1, Figure 2 and Figure 3. All populations showed significant differences (*p* < 0.05) as a function of time. A substantial increase was recorded in the first stage of maturation for both lactic acid bacteria and micrococci. The increase in the two microbial populations is consistent with the literature, which widely recognizes the role of beneficial lactic acid bacteria and CNCs in the ripening of fermented meats [80,81,82,83]. The ready increase in lactic acid bacteria and CNCs can be attributed to the effectiveness of the multi-strain starter culture used. In this regard, the presence of the bacterial strains used as starters was verified. Specifically, typing by the RAPD-PCR of the lactobacilli and CNCs isolated from all batches (G20; G20P; G0 and GP) was performed at 0, 21, and 49 days of maturation. The results demonstrated the efficacy of inoculation and the ability of the strains used as starters to remain dominant throughout the maturation period. More than 90% of the presumptive CNC isolates from the four batches had RAPD-PCR profiles with a similarity coefficient of more than 85% with *S. xylosus* MVS9, and the LAB RAPD-PCR profiles of the LAB isolates were compatible with the *Lt. sakei* 152.

Significant differences were found between batches in the lactic acid bacteria counts depending on the use of glucose. Batches prepared with 0.2% glucose showed significantly higher count levels than those prepared without adding glucose. The increases in the presence of glucose are in line with findings observed by other authors in the presence of glucose [84,85,86]. The level and speed of the growth characterizing lactic acid bacteria in the first weeks of ripening, due to the primary metabolite produced, such as lactic acid, conditions the acidification of the product. Thus, the differences among the samples from the different batches depending on the trend and the level of lactic acid bacteria were also recorded as a pH-dependent variable (Figure 2). The samples of the batches without added sugar showed less acidification; in fact, the pH values found in the samples of batches G0 and G0P were significantly (*p* < 0.01) higher than those found in the samples of the batches with added glucose.

Significant differences were also found among the samples from the different batches with reference to undesirable microorganisms, such as *Enterobacteriaceae* and coliforms (Figure 3). Samples from the batch without sugar and pumpkin oil (G0) showed an initial increase in the enterobacterial and coliform load and a subsequent decrease. Significant differences (*p* < 0.05) from these samples were shown by those from the other batches using glucose alone (G20), in combination with pumpkin oil (G20P), or using pumpkin oil in sugar-free salami (G0P). The decrease in pH, as widely reported by other authors, resulted in a prompt inhibition of undesirable microorganisms. The reduction in enterobacteria and total coliforms was significant, as was the difference between the samples with more acidification and those with milder acidification.

As previously reported in studies conducted on different types of fermented sausages [49,87], the prompt decrease in the pH from the first days of maturation to around 5.2 is a safe condition against total and fecal coliforms, ensuring their effective inhibition. In our study, the trend of *Enterobacteriaceae* and coliforms found in the samples from sugar-free batches is interesting, as they show differences in count levels depending on the use of pumpkin oil. Thus, samples from the G0P batch inhibited enterobacteria and fecal coliforms despite not reaching reassuring acidification levels. This result suggests that, in addition to pH reduction, other mechanisms related to the composition of pumpkin seed oil contribute to antimicrobial effects in fermented salami even when the pH remains above the conventional safety limit. This effect is likely due to the presence of bioactive compounds, unsaturated fatty acids, and antioxidants, which selectively inhibit microbial growth through different mechanisms. In fact, in a previous article, it was pointed out that polyphenols and fatty acids in pumpkin seed oil produce an inhibitory activity against *Enterobacteriaceae* through membrane disruption and enzyme and biofilm inhibition. Water activity (aw) also plays a crucial role in defining the microbial stability of fermented salami. Traditional salami formulations rely on a combination of salt, fermentation, and drying to reduce aw to levels that inhibit pathogenic and spoilage microorganisms [88,89]. Incorporating vegetable oils into fermented salami formulations presents both technological challenges and potential benefits, particularly concerning aw and its impact on product quality. Vegetable oils used in fermented dry sausages, as a substitute for animal fat, may exhibit unusual water retention. Unlike animal fat, which retains water, albeit minimally, vegetable oils in liquid form could affect water activity. Figure 4 shows the significant decrease in aw over time for all samples.

In this study, the use of pumpkin seed oil did not result in significant (*p* > 0.05) differences in the evolution of aw among the diverse sausage batches. This finding indicates that the incorporation of vegetable oils in fermented meat products does not necessarily lead to major variations in aw, despite changes in fat composition. These findings reinforce the notion that vegetable oil incorporation can be a viable strategy to modify the lipid profile of fermented sausages without disrupting key physicochemical properties.

#### 3.2.3. Effect on Lipid Peroxidation and Sensorial Features

The degree of lipid oxidation was assessed by quantifying the level of TBARSs, which, being the second-stage auto-oxidation products obtained from the oxidative cleavage of peroxides into ketones and aldehydes, is a good index for describing the oxidative quality of the products [90,91]. An increase in TBARSs was observed in all samples during ripening. From values between 0.10 and 0.20, they reached levels between 0.26 and 0.8 at the end of the ripening period. Thus, significant differences (*p* < 0.05) were found between the different batches. The most pronounced increase and, consequently, the highest values were found in samples from batch G20. In contrast, the lowest TBARS values, reflecting less oxidation, were found in the samples from batch G0P.

The samples from batches G20P and G0 were in the intermediate position. Therefore, it may be assumed that the final level of TBARSs is influenced by the degree of acidification and the presence of pumpkin oil. Figure 5 shows the TBARS levels based on the multiple regression analysis of several variables, including the level of acidification, lactic acid bacteria load, and oil use. The bubble plot analysis shows that when the level of acidification increases, TBARSs also increase. Therefore, the rate and degree of acidification, whilst playing a crucial role in product safety and stability, may promote the release of reactive oxygen species by accelerating lipid oxidation. These phenomena are even more critical when dealing with buffalo meat, which has unique compositional features compared to pork or beef meat [92]. For instance, buffalo meat has more polyunsaturated fatty acids (PUFAs) than beef and pork, making it more susceptible to lipid oxidation [93,94]. Consequently, a lower pH environment can increase lipid peroxidation, leading to the formation of volatile compounds responsible for rancidity and unpleasant flavors. Moreover, for the oxidation of lipids, it should be added that the rapid oxidation of proteins can be promoted by acidification, leading to the formation of carbonyl compounds, which negatively impact flavor and texture. These oxidative changes can produce bitter, metallic, or stale flavors, ultimately affecting consumer acceptability [95]. This concern is particularly relevant in buffalo meat because of its high heme iron content, which catalyzes oxidative reactions. The results of the present study suggest that a slower, milder acidification process that maintains a relatively higher pH by reducing pro-oxidative conditions may offer significant advantages in improving oxidative stability. In addition, the presence of pumpkin oil contributes to reducing oxidative phenomena. TBARS levels in the samples from the G20P and G0P batches are significantly lower than in the respective oil-free samples (G20 and G0). Other authors have previously reported how different fermentation conditions, such as heat or cold, affect TBARS levels, as well as structure and texture, highlighting the advantage of cold fermentation [90]. In our study, the combination of the two variables, low acidification and the presence of pumpkin oil, allows for the optimal containment of oxidative phenomena, with the lowest levels of TBARSs found. The proper acidification and the choice of specific components can determine the quality characteristics of fermented salami, especially if it is made from buffalo meat. It is reported in the literature that accelerated drying and fermentation processes can cause defects in sensory attributes. Figure 6 shows the scores for the attributes considered for the sensory evaluation.

First, the analysis of the results shows significant differences in relation to color between the samples from the batches with higher acidification (G20 and G20P) compared with those characterized by milder acidification (G0 and G0P). The lower color scores found for the samples from batches G20 and G20P can be attributed to the inherent characteristics of buffalo meat and strong acidification. In fact, it should be considered that buffalo meat has a higher myoglobin content than beef or pork, which leads to a darker red color. The acidification process influences the stability of myoglobin by promoting the formation of nitrosomyoglobin, which is essential for the typical seasoned color of fermented meats. However, excessive acidification or too rapid a decrease in pH can accelerate the oxidation of myoglobin, leading to browning and color deterioration [96]. As expected, the samples from batches G20 and G20P are also characterized by higher pH levels than the samples from the other two batches obtained without the use of sugar. In addition, differences were found in relation to the attribute of “smell”. Samples from lot G20, which were characterized by higher acidification (*p* < 0.05), had lower scores than the others. In this regard, it is interesting to note that the addition of pumpkin seed oil even allowed the odor parameter scores of samples with high acidification (G20P) to be substantially similar (*p* > 0.05) to those recorded for the samples from lots G0 and G0P. Finally, samples from the batches with pumpkin oil additions, regardless of the level of acidification, exhibited higher levels of moisture. This approach not only contributes to a more balanced flavor profile but also helps maintain color and texture, ultimately leading to a higher quality product. Thus, considering the evaluation of the effects on oxidation and the sensory picture, it is possible to conclude that the incorporation of pumpkin seed oil, combined with a milder acidification process during the maturation of dry fermented sausages, improves their overall quality by improving oxidative stability and preserving desirable sensory attributes.

The results obtained pave the way for the prospect of a significant impact on the agri-food sector in areas of southern Italy that are already highly appreciated for the quality of their dairy production (buffalo mozzarella is a cheese that is widely consumed throughout the world). In the case of buffalo meat, the most valuable cuts are eaten as fresh or processed products. Less valuable cuts, on the other hand, are experiencing difficulties finding market outlets. The processing technologies proposed here can therefore act as a driver for the sustainable management of the buffalo supply chain and for the economy of the agro-food sector, enabling small producers to generate significant income. The specific characteristics, with particular attention to the ingredients used, are represented by products with a lower content of saturated fats, with satisfactory health and microbiological profiles, which could be effectively integrated into the Italian gastronomic culture [97].

## 4. Conclusions

The results of this study highlight the crucial role of acidification trends in influencing the quality and stability of fermented salami, particularly in buffalo meat formulations. A slower and milder acidification process demonstrated significant advantages in preserving oxidative stability, contributing to better sensory attribute stability and a more balanced flavor profile. However, acidification and reaching the safe pH threshold are issues to be considered, as some unwanted microorganisms may not be effectively inhibited.

In this regard, this study points out that the incorporation of pumpkin seed oil is a practice that can express antimicrobial and antioxidant properties even in the absence of the pH safety threshold. Overall, this study provides valuable insights for optimizing the fermentation conditions of buffalo meat salami. The combined approach of mild acidification and the strategic use of bioactive vegetable oils offers a promising avenue for improving the sensory attributes, oxidative stability, and microbiological safety of these products.

Future research should explore the long-term effects of these modifications on shelf life and consumer acceptance to further validate their applicability in commercial production.

## Figures and Tables

**Figure 1 foods-14-01077-f001:**
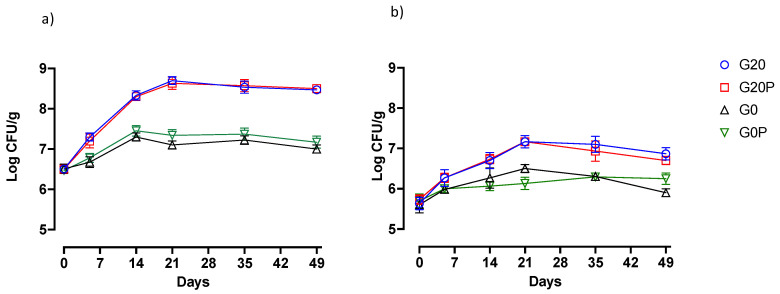
Line graphs showing behaviors of (**a**) lactic acid bacteria and (**b**) Coagulase-Negative Cocci in samples from following batches: G20, fermented sausages prepared with 0.2% glucose; G20P, prepared with 0.2% glucose and pumpkin oil; G0, prepared without glucose; and G0P, prepared with pumpkin oil. Data are reported as mean value with error bars.

**Figure 2 foods-14-01077-f002:**
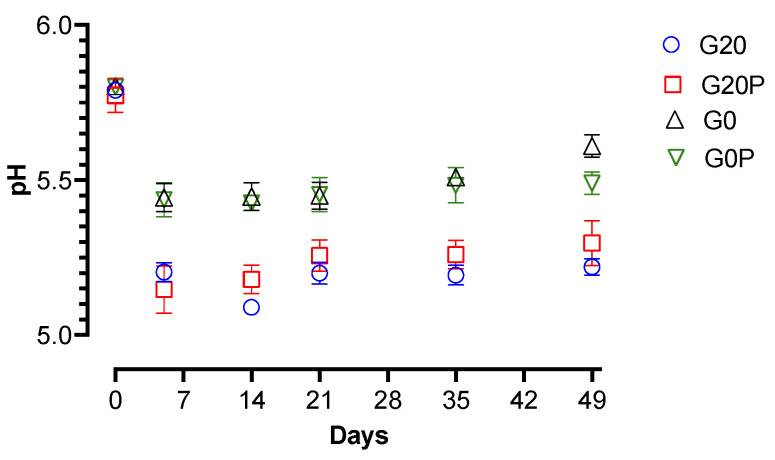
Line graphs show the behaviors of pH in the samples of the batches: G20, fermented sausages prepared with 0.2% glucose; G20P, prepared with 0.2% glucose and pumpkin oil; G0, prepared without glucose; and G0P, prepared with pumpkin oil. The data represented are shown as the mean value with error bars.

**Figure 3 foods-14-01077-f003:**
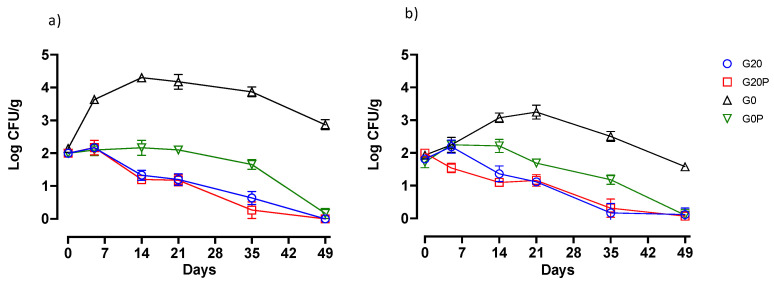
Line graphs show the behaviors of (**a**) *Enterobacteriaceae* and (**b**) total coliforms in the samples of the batches: G20, fermented sausages prepared with 0.2% glucose; G20P, prepared with 0.2% glucose and pumpkin oil; G0, prepared without glucose; and G0P, prepared with pumpkin oil. The data represented are shown as the mean value with error bars.

**Figure 4 foods-14-01077-f004:**
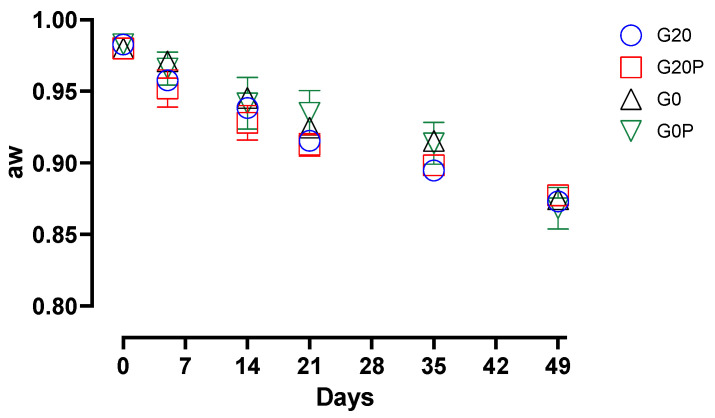
The graph shows the variation in water activity (aw) in the samples of the different batches: G20, fermented sausages prepared with 0.2% glucose; G20P, prepared with 0.2% glucose and pumpkin oil; G0, prepared without glucose; and G0P, prepared with pumpkin oil.

**Figure 5 foods-14-01077-f005:**
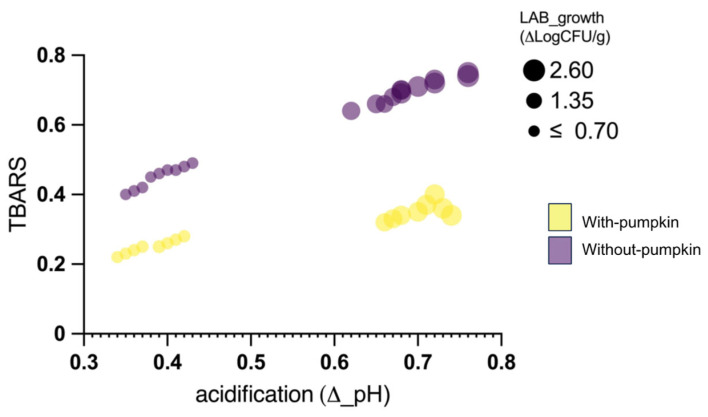
A bubble chart showing the thiobarbituric acid reactive substance (TBARS) values as the multiple linear regression of several variables, including acidification, an increase in LAB count levels, and pumpkin oil addition. The size of the bubbles is directly related to the LAB; the color of the bubbles is related to the presence (yellow) or absence (purple) of oil.

**Figure 6 foods-14-01077-f006:**
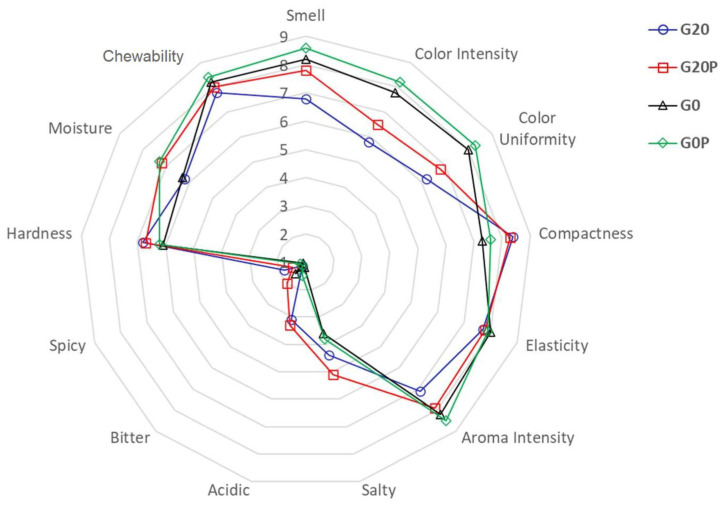
The radar chart shows thirteen sensory attributes at the end of ripening time (49 days) in samples from each batch: G20, fermented sausages prepared with 0.2% glucose; G20P, prepared with 0.2% glucose and pumpkin oil; G0, prepared without glucose; and G0P, prepared with pumpkin oil.

**Table 1 foods-14-01077-t001:** Growth kinetic parameters of *Lt. sakei* 152 and *S. xylosus* MVS9 strains in model food substrate with or without addition (control media) of pumpkin seed oil.

	Control Media	Media with Pumpkin Oil
	*Lt. sakei*	
Initial value	4.05 ± 0.06 ^a^	4.05 ± 0.06 ^a^
Lag	3.85 ± 0.28 ^a^	3.71 ± 0.29 ^a^
Maximum Rate	0.54 ± 0.02 ^a^	0.53 ± 0.02 ^a^
Final Value	9.36 ± 0.06 ^a^	9.27 ± 0.06 ^a^
	*S. xylosus*	
Initial value	4.14 ± 0.08 ^a^	4.06 ± 0.08 ^a^
Lag	4.17 ± 0.5 ^a^	4.11 ± 0.55 ^a^
Maximum Rate	0.35 ± 0.02 ^a^	0.34 ± 0.02 ^a^
Final Value	9.09 ± 0.14 ^a^	9.18 ± 0.14 ^a^

According to *t*-test, values within row with same letter are not significantly different (*p* < 0.05).

**Table 2 foods-14-01077-t002:** Chemical composition in salami at end of maturation from following batches: G20, fermented sausages prepared with 0.2% glucose; G20P, prepared with 0.2% glucose and pumpkin oil; G0, prepared without glucose; and G0P, prepared with pumpkin oil.

	G20	G20P	G0	G0P
Moisture (%)	29.91 ^a^	29.98 ^a^	30.01 ^a^	30.12 ^a^
Protein (% d.m.)	58.50 ^a^	58.58 ^a^	58.63 ^a^	58.97 ^a^
NPN (% d.m.)	10.53 ^a^	10.41 ^a^	10.66 ^a^	10.80 ^a^
Ash (% d.m.)	8.85 ^a^	8.82 ^a^	8.76 ^a^	8.22 ^a^
Carbohydrates (% d.m.)	0.11 ^a^	0.13 ^a^	0.01 ^b^	0.01 ^b^
Lipid (% d.m.)	22.01 ^a^	22.06 ^a^	21.94 ^a^	22.00 ^a^
SFA (% d.m.)	9.94 ^a^	8.01 ^b^	9.81 ^a^	8.05 ^b^
MUFA (% d.m.)	8.66 ^a^	8.44 ^a^	8.49 ^a^	8.52 ^a^
PUFA (% d.m.)	3.41 ^a^	5.61 ^b^	3.64 ^a^	5.43 ^b^
PUFA/SFA	0.34 ^a^	0.70 ^b^	0.37 ^a^	0.67 ^b^

According to the ANOVA statistical test, values within a row with different letters are significantly different (*p* < 0.05).

## Data Availability

The original contributions presented in this study are included in the article/Appendix A. Further inquiries can be directed to the corresponding author.

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
