# Peer review of "Pumpkin Oil and Its Effect on the Quality of Naples-Style Salami Produced from Buffalo Meat"

_foods, 2025, doi:10.3390/foods14061077_

Round 1
Reviewer 1 Report
Comments and Suggestions for Authors
This paper investigated the effect of Pumpkin oil on the the physicochemical properties, nutritional characteristics, microbial safety, sensory quality, and the impact of lipid oxidation in Naples-style salami manufactured with buffalo meat. In my opinion, this work is logical and clear with detailed data analysis. However, the manuscript requires further revisions before it can be considered for publication. The main comments on this work are as follows:
1. Abstract: The statement that buffalo meat possesses "high heme content and specific acidic composition" is mentioned, but the impact of these characteristics on salami production (e.g., microbial activity, or product stability) should be explicitly emphasized in the abstract to better contextualize the research aims.
2. Lines 76-88: The strategy of incorporating bioactive compounds to enhance meat product quality is a well-established concept, while the extensive discussion on non-oil-based bioactive additives appears redundant and tangential to the core focus of this study. This section could be streamlined to prioritize the novel aspects of pumpkin seed oil as the primary bioactive agent under investigation.
3. Line 125: Similarly, the justification for choosing pumpkin seed oil (its antioxidant and antimicrobial properties) needs to be more thoroughly articulated in both the Introduction and Discussion sections.
4. Lines 140-143: Please clarify the origin of the sarcoplasmic proteins (e.g., species-specific source, extraction protocol). Additionally, specify the protein concentration in the medium (mg/mL) and the incubation duration under the described conditions.
5. Line 144: The OD600nm = 0.2 should be correlated with a corresponding viable cell count (CFU/mL) .
6. Table 1: If growth curves of the bacterial strains were measured, including these data either in the main text or as supplemental material would allow readers to assess strain behavior during fermentation.
7. Figure 1: Please ensure consistent font sizes for labels and annotations in fig.1 (a) and (b) to improve visual coherence.
Author Response
Dear Reviewer ,
We sincerely appreciate your thorough evaluation of our manuscript and your constructive feedback. Your insightful comments have helped us refine our work, and we have carefully revised the manuscript accordingly. Below, we address each of your points in detail reporting the answers (A) to each comment (C):
Comment 1 (C 1). Abstract: The statement that buffalo meat possesses "high heme content and specific acidic composition" is mentioned, but the impact of these characteristics on salami production (e.g., microbial activity, or product stability) should be explicitly emphasized in the abstract to better contextualize the research aims.
A1 We revised the abstract to clearly illustrate the reason for the study, the factors analysed and the main results.
In the abstract, we have explicitly pointed out the problems related to the high heme content and specific fatty acid composition of buffalo meat, highlighting how they can encounter sensory and quality decay during the fermentation process by lactic acid bacteria. Therefore, modifications were made as follows “...the high heme content and specific fatty acid composition make the meat particularly susceptible to lactic fermentation, resulting in lipid oxidation and sensory character decay. Therefore, buffalo meat requires tailored fermentation strategies to ensure product stability.” (lines 17 – 20). This revision provides a clearer contextualization of the research aims.
Q2. Lines 76-88: The strategy of incorporating bioactive compounds to enhance meat product quality is a well-established concept, while the extensive discussion on non-oil-based bioactive additives appears redundant and tangential to the core focus of this study. This section could be streamlined to prioritize the novel aspects of pumpkin seed oil as the primary bioactive agent under investigation.
A2. We recognize that the discussion of non-oil-based bioactive additives was overly detailed. As suggested by the reviewer, we have simplified this section to focus more directly on the novel aspects of pumpkin seed oil as the main bioactive agent under study (lines 79-117).
Q3. Line 125: Similarly, the justification for choosing pumpkin seed oil (its antioxidant and antimicrobial properties) needs to be more thoroughly articulated in both the Introduction and Discussion sections.
A3. The justification for choosing pumpkin seed oil has been expanded in both the Introduction (lines: 98-117) and Discussion (lines 480-3486) sections.
Q4. Lines 140-143: Please clarify the origin of the sarcoplasmic proteins (e.g., species-specific source, extraction protocol). Additionally, specify the protein concentration in the medium (mg/mL) and the incubation duration under the described conditions.
A4. We have clarified the origin of sarcoplasmic proteins (lines 143 – 151), specifying the species-specific source (Buffalo Longissimus dorsi) and the extraction protocol. Additionally, we have provided the protein concentration in the medium (10 mg/mL) and detailed the incubation duration (for 30 h at 28 °C)
Q5. Line 144: The OD600nm = 0.2 should be correlated with a corresponding viable cell count of about (CFU/mL).
A5. Line 152-153: The OD600nm = 0.2 has been correlated with a corresponding viable cell count of about 4 log CFU/mL to improve clarity and reproducibility.
Q6. Table 1: If growth curves of the bacterial strains were measured, including these data either in the main text or as supplemental material would allow readers to assess strain behavior during fermentation
A6.Table 1: We appreciate your suggestion and have included bacterial growth curve data as supplemental material (Figure S1) to allow readers to better assess strain behaviour during fermentation.
Q7. Figure 1: Please ensure consistent font sizes for labels and annotations in fig.1 (a) and (b) to improve visual coherence.
A7. Figure 1: The font sizes in Figure 1(a) and (b) have been adjusted for consistency to improve visual coherence and readability.
We are grateful for your valuable input, which has significantly strengthened our manuscript. We hope that the revised version meets the necessary standards for publication.
Reviewer 2 Report
Comments and Suggestions for Authors
The manuscript reported the effect of glucose and pumkimp oil on the fermentation of a salami manufactured with buffalo meat. The novelty of the study is focused on the effect of pumkim oil on the fermentation process in combination with a mild acidification. Auhtors reproted the effect of glucose addition as well although they indicated that this is well known in fermented sausages manufactured with pork meat. Nevertheless, the manuscript showns the intraction between the glucose-fermentation oil and buffalo meat which makes an relevant scientific contibution .
Comments
The abstract needs to be clarified in other to show what has been studied clearly, which factors have been studied? What was the aim of the study?
Line 29 what is a conventional acidification? Please define it or correct the abstract.
Line 129. Please, indicate scientifically how was MIC calculated and add a reference
Line 175. Please define the sampling process. When were analyzed the salamis? How many sausages were analyzed in each batch? How were processed the sausages? Results indicates that several samplings were done at different times of process but this is not reported. Latter in section 2.3.4 it is included the sampling, but this needs to be indicated before to understand what has been done.
Line 194. Please add how the content of fatty acids is expressed.
Line 226. What type of sensory technique was applied? 10 training sessions were performed but then, panellists used a hedonic scale, why? Please provide information about the presentation of samples to the panel.
Line 240. Why an ANOVA analysis was applied to microbial, physical parameters etc? if two different factors, glucose and pumkim oil, have been studied.
Lines 249-250. Acidification and LAB growth is already related, why is this considered individually.
Lines 159 and 393. It would be interesting to also indicate the effect of nitrite as only KNO2 (0.008%) was used. Please justify this low nitrite content. Could nitrite control also enterobacteria?
Line 418-420. Which compounds in oil inhibit entrobacteria? The different mechanism needs to be discussed.
Lines 450-452. These data can not be seen in Tables of figures. Please provide. Also lines 479-480 indicates the TBARS levels in samples but these levels can not be seen.
Figure 3. Please indicate which figure is a and b.
Table 2. Please define how is expressed the content of SFA, UFA, PUFA, and provide definitions in footnote. Is expressed in % of total fatty acids? Or in mg/100g of sausage?
Fig 6. Please explain the difference between smell and Aroma intensity?
M&M section has a high overlap with other authors manuscript, please review it.
Author Response
Dear Reviewer ,
We sincerely appreciate your thorough evaluation of our manuscript and the constructive feedback provided. Below, we address each of your points in detail reporting the answers (A) to each comment (Q):
Comments
Q1. The abstract needs to be clarified in other to show what has been studied clearly, which factors have been studied? What was the aim of the study?
A1. We have revised the abstract to explicitly state the objectives of the study, the factors investigated (glucose addition, pumpkin seed oil inclusion, and their interactions with buffalo meat fermentation), and the key findings (lines 17 -24). This modification ensures clarity regarding the novelty and scope of the research.
Q2. Line 29 what is a conventional acidification? Please define it or correct the abstract.
A2. We have corrected the expression “conventional acidification” in the abstract as follows “acidification in control batch”.
Q3. Line 129. Please, indicate scientifically how was MIC calculated and add a reference
A3. We have better specified the methodology, including the reference (lines 131 -133)
Q4 Line 175. Please define the sampling process. When were analyzed the salamis? How many sausages were analyzed in each batch? How were processed the sausages? Results indicates that several samplings were done at different times of process but this is not reported. Latter in section 2.3.4 it is included the sampling, but this needs to be indicated before to understand what has been done.
A4 We have revised the methodology section detailing the number of samples per batch, processing steps, and time points of analysis (lines 184-186)
Q5. Line 194. Please add how the content of fatty acids is expressed.
A5. Fatty acid content is now explicitly stated as being expressed in percentage of dry matter (g/100 g of dry matter). This clarification has been added to both the text and table
Q6 Line 226. What type of sensory technique was applied? 10 training sessions were performed but then, panellists used a hedonic scale, why? Please provide information about the presentation of samples to the panel.
A6 We have revised this section to provide a clear description of the sensory evaluation methodology. The trained panel was initially calibrated through 10 training sessions using a descriptive analysis approach. The final evaluation is conducted through a descriptive analysis by assessing the perceived intensity of 13 different attributes according to a nine-point scale. Samples from each batch were appropriately sliced and presented individually and anonymously to panelists (judges). The text has been modified lines 232 -248)
Q7. Line 240. Why an ANOVA analysis was applied to microbial, physical parameters etc? if two different factors, glucose and pumkim oil, have been studied.
A7. The statistical approach was clarified and some typos were revised. As already reported in the materials and methods section, the influence (differences and significance levels) of the two growth substrates on the kinetic growth parameters of the microorganisms was assessed by means of t-tests. The typo in the text reporting Anova has been removed and replaced with t-test. While an ANOVA test was applied to assess the individual and interactive effects of glucose and pumpkin seed oil. Data on the development of microbial populations and chemical/physical parameters during the ripening period, as well as sensory attribute scores, were subjected to analysis of variance (ANOVA) followed by a Bonferroni's post hoc test to identify differences and their significance levels (P < 0.05, or P < 0.01) between the different conditions. This review ensures that the statistical analysis is in line with the experimental design.
Q8. Lines 249-250. Acidification and LAB growth is already related, why is this considered individually.
A 8. We sincerely thank the reviewer for his comment. Although acidification is directly related to the growth of lactic acid bacteria (LAB), we considered these parameters separately to better understand their individual contribution to fermentation dynamics. Even if LAB metabolism is the main driver of pH reduction, this relationship can be influenced by multiple formulation factors, including not only the presence or absence of glucose, but also the addition of pumpkin seed oil. Since glucose promotes faster growth and acidification of LAB, while pumpkin seed oil may exert antimicrobial and antioxidant effects, potentially modulating microbial activity, it was essential to evaluate these variables independently. This approach allowed us to distinguish the specific impact of each factor on microbial dynamics, acidification kinetics and overall product stability, ensuring a more complete interpretation of the results.
Q9. Lines 159 and 393. It would be interesting to also indicate the effect of nitrite as only KNO2 (0.008%) was used. Please justify this low nitrite content. Could nitrite control also enterobacteria?
A9. Nitrites were used to control Clostridia and the reduced concentration was in line with the recent European regulation Commission Regulation (EU) 2023/2108 of October 6, 2023.
Q10. Line 418-420. Which compounds in oil inhibit entrobacteria? The different mechanism needs to be discussed.
A10. We have now included a detailed discussion on the antimicrobial activity of pumpkin seed oil, highlighting the role of polyphenols and fatty acids, in inhibiting Enterobacteriaceae via membrane disruption, enzyme and biofilm inhibition. A references was considered (lines 423 – 425) .
Q11. Lines 450-452. These data can not be seen in Tables of figures. Please provide. Also lines 479-480 indicates the TBARS levels in samples but these levels can not be seen.
A11. Figure 4 has been improved. In addition, we have carefully reviewed and ensured that all cited data have been included appropriately. Missing data have been added in a new supplementary table (Table S2).
Q12. Figure 3. Please indicate which figure is a and b.
A12. The figure labels have been corrected to clearly differentiate between (a) and (b), ensuring unambiguous interpretation.
Q13. Table 2. Please define how is expressed the content of SFA, UFA, PUFA, and provide definitions in footnote. Is expressed in % of total fatty acids? Or in mg/100g of sausage?
A13. The table now specifies that SFA, UFA, and PUFA are expressed as percentages of total dry matter (% d.m.) to ensure clarity and consistency.
Q14. Fig 6. Please explain the difference between smell and Aroma intensity?
A14. “Smell” refers to the overall olfactory perception, whereas “aroma intensity” specifically describes the strength of the volatile compounds contributing to the characteristic scent of the product. This distinction has been clarified in the revised manuscript.
Q15. M&M section has a high overlap with other authors manuscript, please review it.
A15. We have carefully reviewed and revised this section to ensure originality while maintaining methodological accuracy. Any unintentional overlap has been corrected.
We appreciate the reviewer’s valuable suggestions, which have significantly improved the clarity and scientific rigor of our manuscript. We hope that the revised version meets the required standards and remains a relevant contribution to the field.
Reviewer 3 Report
Comments and Suggestions for Authors
This is a quality manuscript of very interesting research.
The results in the manuscript build on the results of previous research on fermented meat products, functional ingredients and oxidative stability. However, the use of pumpkin seed oil in dry fermented salami is not often investigated, which adds to the originality of the work. The amount of wording duplication in the manuscript is quite high (percent match: 29% according to the iThenticate report), which should be less.
The introduction is well designed and provides sufficient information on the importance of meat and meat products, as well as their nutritional value and possible health risks. Also, it contains enough information about functional additives in meat products, especially pumpkin seed oil. However, it should be further explained why Napoli-style salami was chosen as a model for this study. How exactly does it fit into this research on fermented meat products? The aim of the research is clearly defined. It would be useful to formulate the hypothesis more precisely in the introduction to make it easier to follow the research.
The main title of the manuscript, according to the above, is too general and short, not the usual form of the main title for this type of scientific publications. I propose corrections and expansion to be specific in the sense of "Assessment of the possibility of production ...".
The Materials and Methods section is described in detail, including the composition and origin of the raw materials used, the method of processing and fermentation of the sausage, as well as analytical methods for chemical, microbiological and sensory analyses. It is not clear on what basis the amount of pumpkin oil (0.5%) added to the sausages was chosen. Also, it should be explained on the basis of which the authors consider it adequate that only 20 subjects participated in the sensory analysis, which is a relatively small number for this type of examination.
It is not entirely clear what the control sample is in the tests. Is it the G20?
The obtained results indicate, and the manuscript provides significant results on the impact of pumpkin seed oil on the quality of Naples-Style Salami salami (as the selected fermented sausage model for testing), a more detailed discussion on possible challenges in industrial production (costs and sustainability of the process), consumer perception, etc. is missing.
Please pay attention to the size and clarity of the graphically displayed data. Match the sizes and layout, for example Figures 1a and 1b with Figure 2, label Figures 3a and 3b, match these figures with, for example Figures 4.
English is at an academic level. There are minor grammatical errors and complex sentences that make it difficult to understand. Additional proofreading would improve the quality of the language.
The manuscript of this paper brings valuable information about the innovative use of pumpkin seed oil in fermented salami, but it would have to be minor improved before publication, with possible additional quality improvement (proofreading) of the English language.
Author Response
Dear Reviewer,
We sincerely appreciate your thoughtful and constructive comments, which have greatly contributed to improving the quality of our manuscript. Below, we address each of your points in detail reporting the answers (A) to each comment (C):
Q1.The introduction is well designed and provides sufficient information on the importance of meat and meat products, as well as their nutritional value and possible health risks. Also, it contains enough information about functional additives in meat products, especially pumpkin seed oil. However, it should be further explained why Napoli-style salami was chosen as a model for this study. How exactly does it fit into this research on fermented meat products? The aim of the research is clearly defined. It would be useful to formulate the hypothesis more precisely in the introduction to make it easier to follow the research.
A1. We have expanded the Introduction (Lines 79 -118) to clarify the rationale for selecting Naples-style salami as the model for this study. This traditional dry-fermented sausage, known for its characteristic physicochemical properties and production process, provided an ideal framework to assess the impact of pumpkin seed oil on lipid oxidation, microbial stability, and sensory quality. Additionally, we have refined the research hypothesis to enhance clarity and coherence.
As reported at lines 104 -115 Salami style Naples, a cornerstone of the delicatessen of the geographical area with a strong buffalo vocation, would represent a valid model for the use of buffalo meat and the experimentation of new ingredients and fermentation processes. Specifically, Napo-li-type salami is produced in Campania, the cradle of buffalo breeding and production. While the most prized parts of the buffalo easily find a market, the less renowned cuts, such as the shoulder, can be used in the innovative production of traditional cured meats, thus contributing to the preservation and renewal of local gastronomic traditions, with a positive impact on the economy and sustainability of production. This type of salami is characterized by a finely balanced mixture of meat, fat and other ingredients, subjected to careful fermentation and drying process.
Additionally, we have refined the research hypothesis to enhance clarity and coherence.
Q2. The main title of the manuscript, according to the above, is too general and short, not the usual form of the main title for this type of scientific publications. I propose corrections and expansion to be specific in the sense of "Assessment of the possibility of production ...".
A2 In line with your suggestion, we have revised the manuscript title to make it more specific and reflective of the study’s scope, explicitly mentioning the effects of pumpkin seed oil in the production of Naples-style salami. So, the new title is “Use of Pumpkin Oil to Assess it Effect on the Quality of Naples-Style Salami Produced From Buffalo Meat”
Q3.The Materials and Methods section is described in detail, including the composition and origin of the raw materials used, the method of processing and fermentation of the sausage, as well as analytical methods for chemical, microbiological and sensory analyses. It is not clear on what basis the amount of pumpkin oil.
A3. We have now explicitly stated the rationale behind the selected concentration of pumpkin seed oil, which was based on its antimicrobial effects vs undesirable and useful microorganisms (lines 162-163).
Q4. Also, it should be explained on the basis of which the authors consider it adequate that only 20 subjects participated in the sensory analysis, which is a relatively small number for this type of examination.
A4. We acknowledge that the sensory analysis was intentionally conducted with a trained panel of (n=20) judges from 40 volunteers. This choice was in line with established practices in descriptive sensory analysis, which as also reported in the literature (DOI: 10.1111/jtxs.12616), a trained sensory panel typically consists of about 12 evaluators, with panel sizes ranging from 10 to 20 judges commonly accepted for structured sensory evaluations. In our study, panelists underwent rigorous selection and training to ensure consistency, reliability and sensitivity in evaluating the specific sensory attributes of buffalo meat salami. These aspects have also been reported in the text under the material and methods section (lines 235 – 251)
Q5. It is not entirely clear what the control sample is in the tests. Is it the G20?
The control sample in the study corresponds to the G20 formulation. We have now clarified this point in the Materials and Methods section to avoid any ambiguity.
Q6.The obtained results indicate, and the manuscript provides significant results on the impact of pumpkin seed oil on the quality of Naples-Style Salami salami (as the selected fermented sausage model for testing), a more detailed discussion on possible challenges in industrial production (costs and sustainability of the process), consumer perception, etc. is missing.
We appreciate this insightful comment and we have expanded the Discussion section (lines 527 -537)
Q7.Please pay attention to the size and clarity of the graphically displayed data. Match the sizes and layout, for example Figures 1a and 1b with Figure 2, label Figures 3a and 3b, match these figures with, for example Figures 4.
We have revised all figures to ensure consistency in size, layout, and labeling, particularly aligning Figures 1a, 1b, and Figure 2, as well as properly labeling Figures 3a and 3b to match the format of Figures 4.
Q8.English is at an academic level. There are minor grammatical errors and complex sentences that make it difficult to understand. Additional proofreading would improve the quality of the language.
To enhance readability and academic rigor, we have carefully revised the manuscript, correcting grammatical errors and simplifying complex sentences.
Q9. The manuscript of this paper brings valuable information about the innovative use of pumpkin seed oil in fermented salami, but it would have to be minor improved before publication, with possible additional quality improvement (proofreading) of the English language.
A9. We have sought proofreading to ensure linguistic accuracy.
Once again, we thank you for your valuable feedback, which has significantly strengthened our work. We hope that our revisions adequately address all concerns and that the improved manuscript meets the journal’s standards for publication.
Best regards
SJL
Reviewer 4 Report
Comments and Suggestions for Authors
The manuscript comprises a novel in-nature research study that uses pumpkin seed oil to monitor the freshness and overall quality of fermented sausage prepared from buffalo meat. The authors run physicochemical, microbiological, and sensory analyses to indicate the differences in the product's composition during storage. The paper falls deeply within the aims and scope of the journal.
The manuscript has been well prepared and provides detailed data treated with statistical analysis. The figures and tables are in good format and easy to read. The attached PDF suggests some minor improvements for the authors.
Based on the overall quality of this study, I suggest a minor revision.

The English language is fine, but minor improvements are required.
Author Response
Dear Reviewer,
I would like to express my sincere thanks for your helpful suggestions to improve the clarity and accuracy of the work. We have incorporated your comments into the text. We are grateful for the time you took to review our manuscript. We thank you again for your support and commitment.
Sincerely yours,
Silvia Jane Lombardi
Round 2
Reviewer 1 Report
Comments and Suggestions for Authors
the paper may be accepted and pulished.